# Risk stratification for hospital-acquired venous thromboembolism in medical patients (RISE): Protocol for a prospective cohort study

Damien Choffat[1]*, Pauline Darbellay Farhoumand[2], Evrim Jaccard[1], Roxane de la Harpe[1], Vanessa Kraege[1], Malik Benmachiche[1], Christel Gerber[1], Salomé Leuzinger[3], Clara Podmore[3], Minh Khoa Truong[4], Céline Dumans-Louis[1], Christophe Marti[2], Jean-Luc Reny[2], Drahomir Aujesky[5], Damiana Rakovic[5], Andreas Limacher[6], Jean-Benoît Rossel[3,6], Christine Baumgartner[5‡], Marie Méan[1‡]

1 Division of Internal Medicine, Department of Medicine, Lausanne University Hospital (CHUV), Lausanne, Switzerland, 2 Division of General Internal Medicine, Department of Medicine, Geneva University Hospitals (HUG), Geneva, Switzerland, 3 Center for Primary Care and Public Health (Unisanté), University of Lausanne, Lausanne, Switzerland, 4 Division of Pneumology, Department of Medicine, Lausanne University Hospital (CHUV), Lausanne, Switzerland, 5 Department of General Internal Medicine, Inselspital, Bern University Hospital, University of Bern, Bern, Switzerland, 6 CTU Bern, University of Bern, Bern, Switzerland

‡ CB and MM are Joint Senior Authors
* Damien.choffat@chuv.ch

**Data Availability Statement:** No datasets were generated or analysed during the current study. All relevant data from this study will be made available

## Abstract

### Background

Hospital-acquired venous thromboembolism (VTE) is one of the leading preventable causes of in-hospital mortality. However, its risk assessment in medically ill inpatients is complicated due to the patients' heterogeneity and complexity of currently available risk assessment models (RAMs). The simplified Geneva score provides simplicity but has not yet been prospectively validated. Immobility is an important predictor for VTE in RAMs, but its definition is inconsistent and based on subjective assessment by nurses or physicians. In this study, we aim to prospectively validate the simplified Geneva score and to examine the predictive performance of a novel and objective definition of in-hospital immobilization using accelerometry.

### Methods and analysis

RISE is a multicenter prospective cohort study. The goal is to recruit 1350 adult inpatients admitted for medical illness in three Swiss tertiary care hospitals. We collect data on demographics, comorbidities, VTE risk and thromboprophylaxis. Mobility from admission to discharge is objectively measured using a wrist-worn accelerometer. Participants are followed for 90 days for the occurrence of symptomatic VTE (primary outcome). Secondary outcomes are the occurrence of clinically relevant bleeding, and mortality. The evolution of autonomy in the activities of daily living, the length of stay, and the occurrence of readmission are also recorded. Time-dependent area under the curve, sensitivity, specificity, and positive and negative predictive values are calculated for each RAM (i.e. the simplified and

upon study completion. Deidentified research data will be made publicly available when the study is completed and published.

**Funding:** The RISE cohort is funded by several non-profit foundations (SGAIM Foundation, Novartis Biomedical Research Foundation, Swiss Heart Foundation, Chuard Schmidt Foundation, Gottfried und Julia Bangerter Foundation). The funding sources had no role in study design; in the collection, analysis and interpretation of data; in the writing of the report; and in the decision to submit the article for publication.

**Competing interests:** The authors have declared that no competing interests exist.

original Geneva score, Padua, and IMPROVE score) with and without the objective mobility measures to assess their accuracy in predicting hospital-acquired VTE at 90 days.

## Ethics and expected impact

The ethics committee approved the protocol and the study was registered on ClinicalTrials. gov as NCT04439383. RISE has the potential to optimize VTE risk stratification, and thus to improve the quality of care of medically hospitalized patients.

## Introduction

Hospital acquired venous thromboembolism (VTE), defined as pulmonary embolism (PE) or deep vein thrombosis (DVT), is one of the leading preventable causes of in-hospital mortality [1]. About 75% of all hospital-acquired VTE occur in hospitalized medical patients [2]. So much so that hospitalization for an acute medical illness is per se a risk factor for VTE [3].

Randomized-controlled trials (RCTs) conducted 15 to 20 years ago showed significant reductions in VTE with the use of heparin compared to placebo in selected medical inpatients [4–6]. However, pharmacological VTE prophylaxis increases the risk of bleeding [4]. Guidelines recommend providing pharmacological thromboprophylaxis (TPX) to hospitalized medical patients only if they are *at increased risk of VTE* during their hospital stay [7,8].

Assessing thromboembolic risk in medical inpatients is currently done empirically or using risk assessment models (RAMs) incorporating an array of demographic and clinical patient characteristics. Available validated RAMs, such as the original Geneva score [9], the Padua [10] or the IMPROVE score [11,12], have various shortcomings including a suboptimal sensitivity to identify high VTE risk patients (ranging from 73% to 90% among any of the RAMs) [13]. Furthermore, they have a large number of items score, some of which are not available at admission (e.g. ICU stay) [12]. The simplified Geneva score has recently been developed as a simpler and more usable RAM [13]. Prospective validation is needed before it can be implemented in everyday clinical practice. To that end, the first aim of this study is to externally validate this novel RAM.

Being an important risk factor for hospital-acquired VTE, immobilization is included in existing RAMs [9,10,12,14]. However, due to the lack of a standardized definition, its usefulness is limited [13,15]. In everyday practice, the degree of immobilization is estimated subjectively, based either on the physician's own perception or on nursing assessment [16–21], with a questionable accuracy [21]. Patients and hospital staff also interpret physicians' orders of mobilization with a substantial variation; for example ambulation orders "out of bed to chair" can lead to a daily step count of 0 to 1800 (0–1.3 km) [21].

Recent evidence suggests that objective measures of mobility using a wrist-worn tri-axis accelerometer improves the accuracy of mobility assessment in hospitalized patients [22–26]. Whether objective mobility measures could predict hospital-acquired VTE, and whether incorporation of these measures into VTE RAMs could improve their predictive ability has yet to be examined. Therefore, we aim to establish the predictive performance of a novel and objective definition of in-hospital immobilization using accelerometry.

Overall, risk assessment and prevention of hospital-acquired VTE remains a major challenge for hospital physicians, and expert societies have called for further research on this topic [8]. To that end, this prospective cohort study aims to improve VTE prevention in hospitalized medical patients.

## Objectives and hypotheses

The primary objective is to prospectively validate the simplified Geneva score and to compare its prognostic performance with previously validated RAMs (i.e., the original Geneva, Padua, and IMPROVE scores). Therefore, we hypothesize that the novel, easier-to-use simplified Geneva score will be able to accurately detect medical inpatients at risk of hospital-acquired VTE and that it will be at least as accurate as previously validated RAMs.

Our second objective is to develop a new, objective, definition of inpatient immobilization using accelerometry and to compare its performance in predicting hospital-acquired VTE with that of the subjective measurement. Accordingly, we hypothesize that objective, accelerometry-assessed mobility will be more accurate in predicting the risk of hospital-acquired VTE than subjective physician perception and that its incorporation into the simplified Geneva score will improve its prognostic performance.

## Materials and methods

### Study design and setting

RISE (RIsk Stratification for hospital-acquired venous thromboEmbolism in medical patients) is a multicenter prospective cohort study including consecutive consenting adult patients admitted to the general internal medicine wards of three Swiss university hospitals (i.e., the Lausanne, Bern, and Geneva). The recruitment started June 22, 2020 and we expect that the last participant will finish the study in Spring 2022.

### Patient selection

Consecutive adult patients with acute illness admitted for more than 24 hours to a general internal medicine ward are eligible. Exclusion criteria are the need for therapeutic anticoagulation (e.g., due to atrial fibrillation), estimated life expectancy of less than 30 days, insufficient proficiency of the German or French language, or prior enrolment in the cohort.

Importantly, patients with mental illness or cognitive impairment are not excluded from the study. Indeed, these disorders are frequently encountered in older patients, whose risk of VTE and immobilization are particularly high [27,28].

### Ethical aspects

This study is conducted in accordance with the Declaration of Helsinki, the ICH-GCP guidelines, and all applicable legal/regulatory requirements. The Ethics Committee of the Canton of Berne (Kantonale Ethikkommission für die Forschung, Kanton Bern) authorized the RISE study on (Reference number: 2020–00606).

### Baseline data collection and VTE risk assessment

For all eligible and consenting participants, study personnel prospectively collect demographic data (sex, year of birth, body weight, height, setting prior to admission), information on comorbidities (including all items of the Charlson Comorbidity Index [29,30]), medications at admission with a potential antithrombotic effect (aspirin, other antiplatelet therapy, nonsteroidal anti-inflammatory drugs), potential contraindications to pharmacological VTE prophylaxis (known hypersensitivity to heparin and history of heparin induced thrombocytopenia), and laboratory variables (thrombocytopenia, spontaneous international normalized ratio > 2 (INR), kidney failure and anemia) known to affect pharmacological TPX provision (Table 1).

At baseline, all items of the simplified and original Geneva score, the IMPROVE score, and the Padua score are collected (Table 2), and the score for each RAM is calculated in order to

**Table 1. Baseline data collection.**

**Demographic characteristics**
Sex, year of birth, date of admission, date of study inclusion, body weight (kg), height (cm), setting prior to admission

**Items of the risk assessment models**
Previous VTE, hypercoagulable state/thrombophilia, active cancer, history of cancer within last 5 years, myeloproliferative syndrome, cardiac failure, respiratory failure, acute infection, rheumatologic disorder, immobilization (bed rest with bathroom privileges) ≥72 hours, estimated immobilization >7d, stroke (and date of event), myocardial infarction (and date of event), recent (≤1 month) trauma or surgery (and date of event), ongoing hormonal treatment, lower extremity paralysis/paresis, stay in the intensive care unit / intermediate care unit, nephrotic syndrome, recent travel (>6 hours), chronic venous insufficiency, pregnancy, dehydration

**Comorbidities [29]**
History of myocardial infarction, congestive heart failure, peripheral vascular disease, cerebrovascular disease, dementia, chronic obstructive pulmonary disease, connective tissue disease, peptic ulcer disease, liver disease, diabetes mellitus, hemiplegia, chronic kidney disease, localized solid tumor, metastatic solid tumor, leukemia, lymphoma, AIDS, gastroduodenal ulcer, history of bleeding, inflammatory bowel disease; number of comorbidities

**Contraindications to pharmacological VTE prophylaxis**
Known hypersensitivity to heparin, history of heparin induced thrombocytopenia, liver failure, active non-major or major bleeding (and date of event), hemorrhagic transformation of acute ischemic stroke (and date of event)

**Laboratory findings**
Platelet count, international normalized ratio, serum creatinine, hemoglobin

**Medications at admission**
aspirin, other antiplatelet therapy (clopidogrel, prasugrel, ticagrelor), nonsteroidal anti-inflammatory drugs

**Autonomy**
Modified Barthel Index; Braden scale; location of eating, eliminating urine or stool, and washing

Abbreviation: VTE, venous thromboembolism; d, days.

categorize study participants into risk groups for VTE. All demographic, clinical, and laboratory data are collected from electronic health records (EHR) or at the patient's bedside (from the patient and/or nurse in charge) by trained study personnel.

Treatments during hospital stay affecting the risk of hospital-acquired VTE or bleeding are recorded from the EHR, including type and duration of the pharmacological (low-molecular-weight heparin, unfractionated heparin, fondaparinux, other) and mechanical TPX (lower extremity compression stockings/bandages, intermittent pneumatic compression devices). In case therapeutic anticoagulation is initiated, the start date and the indication are documented. Furthermore, information on red blood cell transfusions, central venous catheter, and surgical procedures during hospitalization are recorded [32,33].

Autonomy in the activities of daily living (ADL) prior to hospitalization is assessed at admission using the modified Barthel Index [34]. For patients with cognitive impairment or confusion, the level of ADL autonomy is assessed by interviewing their relatives or caregivers. The modified Barthel Index has been reported as being the most accurate scale to assess activities of daily living (ADL) and has thus been widely used as a measure of autonomy [34]. The patient's ability to perform different ADLs is rated as follows: fully independent, with minimal or moderate help, attempts task but putting him/herself at risk, or unable to perform. The maximum point score is 100; a total modified Barthel Index point score of 0–20 suggests total, 21–60 severe, 61–90 moderate, and 91–99 slight dependence. A point score of 100 indicates that the patient is independent of assistance from others.

## Mobility assessment

Objective measurement of mobility is done with a wrist-worn tri-axis accelerometer (GENEActiv Original, ActivInsights Ltd, UK, https://www.activinsights.com/actigraphy/geneactiv-original/),

**Table 2. VTE risk assessment models for risk stratification in hospitalized medical patients.**

| Score Items | Points | | | |
|---|---|---|---|---|
| | Simplified Geneva Score [13] | Original Geneva Score [9] | Padua Score [10] | IMPROVE Score [12,14] |
| Previous VTE | 3 | 2 | 3 | 3 |
| Hypercoagulable state [a] | 2 | 2 | 3 | 2 |
| Cancer [b] [9,10,31] | 2 | 2 | 3 | 2 |
| Myeloproliferative syndrome [c] | | 2 | | |
| Cardiac failure [d] | 2 | 2 | 1 | |
| Respiratory failure [e] | | 2 | | |
| Acute infection | 2 | 2 | 1 | |
| Acute rheumatologic disorder [f] | | 2 | | |
| Immobilization | 2[g] | 1[g] | | 1[h] |
| Reduced mobility | | | 3[i] | |
| Lower limb paralysis or paresis [31] | | | | 2 |
| Age >60 years | 1 | 1 | | 1 |
| Age >70 years | | | 1 | |
| Body mass index $\geq 30 kg/m^2$ | 1 | 1 | 1 | |
| Recent stroke ($\leq$ 3 months) [9] | 1 | 2 | 1 | |
| Recent myocardial infarction ($\leq$ 1 month) [9] | | 2 | | |
| Nephrotic syndrome | | 2 | | |
| Hormonal treatment [j] | | 1 | 1 | |
| Travel within last 7 days (>6 hours) | | 1 | | |
| Chronic venous insufficiency | | 1 | | |
| Pregnancy | | 1 | | |
| Dehydration | | 1 | | |
| Recent trauma or surgery (<1 month) | | | 2 | |
| Stay in intensive or coronary care unit | | | | 1 |
| **Cut-offs [8–10,12,13]** | | | | |
| Low VTE risk | 0–2 | 0–2 | 0–3 | 0–1 |
| High VTE risk | $\geq$3 | $\geq$3 | $\geq$4 | $\geq$2 |

Abbreviations: VTE, venous thromboembolism.

[a] anti-thrombin deficiency, APC resistance, protein C or protein S deficiency, factor V Leiden, G20210A prothrombin-mutation, antiphospholipid syndrome.

[b] metastatic cancer, or cancer treated with radiotherapy/chemotherapy/immunotherapy, or cancer surgery within last 6 months (also relates to myeloma or myelodysplastic syndrome), excluding non-melanoma skin cancer.

[c] essential thrombocytopenia, polycythemia vera, primary myelofibrosis, chronic myeloic leukemia.

[d] acute or chronic heart failure of any cause with a preserved or reduced ejection fraction.

[e] acute or chronic need for supplemental oxygen.

[f] rheumatoid arthritis, vasculitis, or connective tissue disease.

[g] immobilization was defined as complete bedrest or inability to walk for >30min per day for $\geq$3 days [9].

[h] immobilization was considered if the patient was being confined to bed or chair with or without bathroom privileges for $\geq$7 days immediately prior to and during hospital admission [31].

[i] reduced mobility was defined as anticipated bed rest with bathroom privileges for $\geq$3 days [10].

[j] contraception, post-menopausal hormone therapy, antitumor therapy containing estrogen, ethinylestradion, estradiol.

parametrized at 50 Hz. The accelerometer is provided to patients immediately after inclusion. Patients are asked to wear the device continuously (day and night, including while showering) until hospital discharge or transfer to another department (e.g., intensive care, surgery unit, etc.). Accelerometry data is extracted and analyzed using the GGIR package for R (version 1.11 or later) [35]. A valid day of mobility measurement is defined as at least 10 hours of wearing the

accelerometer during daytime, and at least 24 hours of valid data is required for analysis [36,37]. In the analysis, we consider the following measurements: minutes per day in different types of activities, no activity or sleep; total minutes during a day spent active/inactive; mean acceleration in miliG/vector. Physical activity raw data is further processed using the Verisense Step Count Algorithm for GGIR (https://github.com/ShimmerEngineering/Verisense-Toolbox/tree/master/Verisense_step_algorithm) and the open source GENEAclassify R-package (https://cran.r-project.org/web/packages/GENEAclassify/GENEAclassify.pdf), in order to obtain the number of steps taken per day. We estimate the percentage of time of a patient's mobility, using a cut-off of <4 steps and ≥4 steps taken per minute to define periods of immobilization and mobilization, respectively, as previously reported in a study of medical inpatients [24].

For the subjective mobility measurement, we consider the patient's, the nurse's and the hospital treating physician's mobility estimates. Patients are asked about their ability to walk, i.e. whether they are able to walk independently, with assistance from one or two people, with or without mobility aids, or if they are unable to walk at all. Furthermore, they are asked about the location of eating (bed, edge of bed, table), urinating and defecating (bed, chair next to bed, bathroom), and washing (bed, chair in front of sink, shower).

Nurses' assessment of mobility is performed through items of the Braden scale [38]. The Braden scale has been developed and validated to identify hospitalized patients at risk of pressure sores. This scoring system includes six items with a total score ranging from 0 to 23. Patients with a score of nine or less are categorized as having a very high risk. Two items of this score are specifically dedicated to physical activity: "degree of physical activity" (patient is bedfast, chairfast, walks occasionally, walks often) and "ability to change and control position" (patient is completely immobile, very limited, slightly limited, or has no limitation in mobility). Therefore, nurses indirectly assess the mobility of patient.

On the second day of hospitalization, the physician is asked whether a corresponding patient fulfills the different immobilization criteria as defined in each RAM (Table 2). The physician is also asked to subjectively estimate the patient's ability to ambulate in standardized terms (i.e., no ambulation, out of bed to chair, out of bed to ambulate once daily, twice or 3 times daily, or ambulate ad libitum) [21]. Hospital treating physicians are contacted on the second day of hospitalization rather than on admission because the decision to prescribe TPX is most likely already made and thus unlikely to be influenced by questions on the patient's mobility status. Finally, information on physical therapy orders for mobilization are collected from EHR. The prescriptions of specific ambulation regimens or physical therapy are left at the discretion of the hospital treating physician.

## Primary and secondary outcomes

The primary outcome is symptomatic objectively confirmed fatal and non-fatal VTE, defined as distal or proximal DVT or PE up to 90 days after hospital admission. As described in previous studies, the objective diagnostic of PE is based on available radiology (CT pulmonary angiography, pulmonary angiography, or ventilation-perfusion lung scan) or autopsy reports [39–42] (S4 File). Likewise, the objective diagnostic of DVT is based on compression ultrasonography or contrast venography [39,43]. As only symptomatic VTE events are recorded, patient need to present symptoms such as dyspnea, couch, acute chest pain or syncope for PE, and unilateral pain or swelling or erythema for DVT [44]. VTE events diagnosed during the first 48 hours of hospitalization are not considered as a primary outcome for this study in order to rule out pre-existing VTE that occurred prior to hospital admission [45]. In line with previous studies on VTE prophylaxis and given similarities in some risk factors and outcomes [31,46], symptomatic upper extremity DVT is also considered as a study outcome, although its incidence is expected to be low [46].

Secondary medical outcomes are the occurrence of major bleeding, clinically relevant non-major bleeding, and all-cause mortality during the follow-up period. As described in two previous Swiss VTE studies, SAFE-SSPE [40] and SWITCO65+ [43,47], major bleeding is defined as fatal bleeding, symptomatic bleeding at critical sites, bleeding with a reduction of hemoglobin of at least 20 g/L or bleeding leading to transfusion of 2 or more units of packed red blood cells [48]. Likewise, clinically relevant non-major bleeding is defined as overt bleeding that does not meet criteria for major bleeding but is associated with a medical intervention, unscheduled physician contact (visit or telephone call), or pain, or impairment of activities of daily life [49]. All-cause mortality is categorized as PE-related, bleeding-related, due to another cause or due to an undetermined cause according to already published criteria [40,50–53]. In addition, the evolution of the autonomy in the ADL during the follow-up period, using the modified Barthel Index, the length of hospital stay and rehospitalization for an acute medical illness up to 90 days are also considered as secondary outcomes.

All medical outcome events (hospital-acquired VTE, major and clinically relevant non-major bleeding, and death) are reviewed and adjudicated by a committee of three independent clinical experts. The final adjudication is based on the committee's full consensus.

## Study procedures

Study investigators screen consecutive patients newly admitted to general internal medicine of participating clinics for eligibility on weekdays (Figs 1 and 2). Eligible patients are informed about the study aims/procedures and asked to provide written informed consent. For patients who are unable to give informed consent due to mental illness or cognitive impairment, permission to participate in the study is obtained from a legally authorized representative. Participating patients are equipped with an accelerometer for collection of mobility data throughout the hospital stay, and trained study personnel collect patient baseline data on the day of enrolment (Fig 1).

A follow-up visit is conducted prior to discharge to collect information on discharge location, information on treatments since admission with a focus on pharmacological and mechanical TPX, patient autonomy, and clinical outcomes (Fig 1). Study investigators collect the accelerometer and upload accelerometry data to the database using the relevant software.

A follow-up phone call is performed at day 90 ± 5 after study inclusion by trained study personnel. In case of unavailability of the patient, their designated contact person or general practitioner is called instead. Information on outcomes is assessed. As initiation of therapeutic anticoagulation during follow-up affects the outcomes, we also collect information about the potential introduction of therapeutic anticoagulation since discharge (Fig 1). In case of the occurrence of a medical outcome event, study personnel collect all available documentation (e.g. medical reports, laboratory and imaging data) related to the event for the adjudication process.

## Sample size calculation

We performed the sample size calculation for the primary objective, i.e. the validation of the simplified Geneva score for the prediction of hospital-acquired VTE. Based on a 2010 Swiss cohort study [13], we assume that that 67% of patients are categorized as high risk and 33% as low risk based on the simplified Geneva score, and a 90-day incidence of hospital-acquired VTE with adequate thromboprophylaxis of 2.8% and 0.6%, in high risk and low risk patients, respectively, according to the simplified Geneva score [13]. Therefore, we will need 1308 patients to detect an absolute risk difference of 2.2% between high and low risk patients, with a power of 80% at a 2-sided alpha of 0.05. The numbers stated above correspond to a relative risk of 4.7, a sensitivity of 90%, and a specificity of 34% [13]. This sample size provides

| | Screening | Enrolment & Baseline | | Follow-up | Close-out |
|---|---|---|---|---|---|
| **Study Period** | Screening | 1 | | 2 | 3 |
| **Visit** | Screening | 1 | | 2 | 3 |
| **Time-point** | Hospital admission, d0 | | Day after inclusion, d1 | Day of/prior to discharge | End of follow-up, d90 |
| **ENROLMENT:** | | | | | |
| Eligibility screen | X | | | | |
| Informed consent | | X | | | |
| **INTERVENTION:** | | | | | |
| Accelerometry | | ← | —————————————— | → | |
| **ASSESSMENTS:** | | | | | |
| **Baseline characteristics** | | | | | |
| Demographic characteristics | | X | | | |
| Items of the RAMs | | X | | | |
| Comorbidities | | X | | | |
| Contraindications to pharmacological VTE prophylaxis | | X | | | |
| Laboratory findings | | X | | | |
| Medications at admission | | X | | | |
| Treatments since admission | | X | | X | |
| Modified Barthel Index | | X | | X | X |
| Ability to walk (patient's assessment) | | X | | | |
| Ability to walk (physician's assessment) | | | X | | |
| **Medical Outcomes** | | | | | |
| Symptomatic VTE | | | | X | X |
| All-cause mortality | | | | X | X |
| Major & clinically relevant non-major bleeding | | | | X | X |
| **Other study data** | | | | | |
| Length of hospital stay | | | | X | |
| Discharge location | | | | X | |
| Subsequent hospitalizations | | | | | X |

**Fig 1. Timeline of patient enrolment and schedule of data collection.** Adapted from the SPIRIT statement [54]. Abbreviations: d, day; RAM, risk assessment model; VTE, venous thromboembolism.

sufficient precision for the validation of the simplified Geneva score. Assuming an area under the curve (AUC) of 0.75, the normal-approximation 95% confidence interval (CI) ranges from 0.64 to 0.86. For sensitivity and specificity, the 95% Wilson CIs range from 72% to 96% and from 31% to 36%, respectively. We will recruit a total sample of 1350 patients to account for potential dropouts, which we expect to be few given the low follow-up burden [10].

For the second objective, namely the assessment of objective mobility measurement to predict the risk of VTE, the same measures of association and prognostic accuracy as described above are estimated. The sample size of 1350 patients provides comparable precision as stated above.

## Planned statistical analyses

Once the 1350 patients will have completed the study, the following statistical analyses will be conducted. First, time to event analyses with competing risk methods will be used to assess the prognostic performance of the simplified Geneva score and the other RAMs (Table 2) and

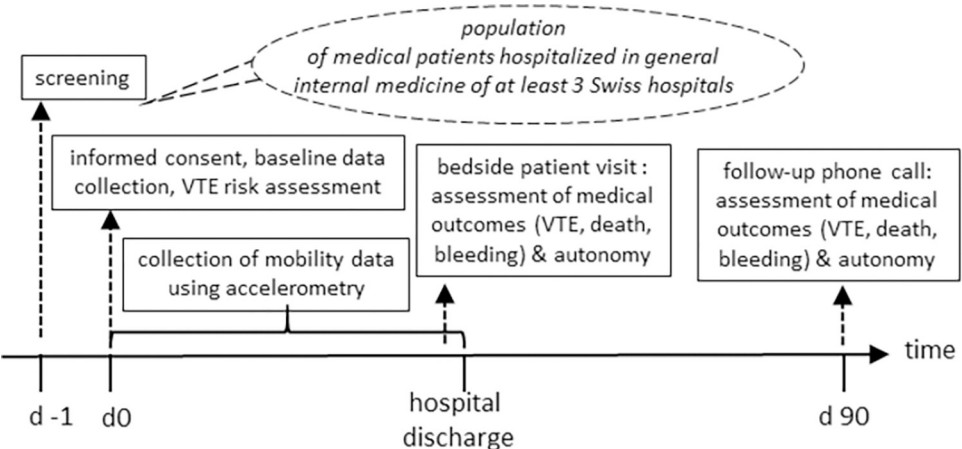

**Fig 2. Study organization and follow-up.**

their association with hospital-acquired VTE, with non-VTE death representing the competing risk. We will use a subdistribution hazard model of Fine and Gray [55] to assess the association of the simplified Geneva score and the other RAMs with VTE, calculating subhazard ratios with 95% CIs. These analyses will be adjusted for the use of TPX and study site. Cumulative incidences of hospital-acquired VTE in low- and high-risk score patients will be assessed and graphically presented to assess calibration and compare different RAMs. The time-dependent AUC as well as sensitivity, specificity, and positive and negative predictive values will be calculated for each RAM to assess their accuracy to predict hospital-acquired VTE at 90 days using time-dependent receiver operating characteristic (ROC)-curve analysis, taking into account censored data and competing events.

Additionally, as some patients are treated with TPX, we will perform separate sensitivity analyses in patients with and without TPX. It is possible that during follow-up, a small number of patients will be started on therapeutic dose anticoagulation for reasons other than VTE (e.g., new onset atrial fibrillation); data of these patients will be censored in the main analysis.

Secondary time-to-event outcomes (major and clinically relevant non-major bleedings) will also be evaluated using competing risk regression. For all-cause mortality, we will use an ordinary Cox regression, for length of stay an accelerated failure time model. Binary outcomes (in-hospital VTE and readmission) will be evaluated using logistic regression. The Barthel-index will be assessed using linear regression. All models will be adjusted for the use of TPX and study site.

We will examine the association between subjective (physician's perception) and objective (accelerometry-measured) mobility levels, as a continuous measure as well as divided into quartiles, and 90-day cumulative incidences of HA-VTE using competing risk regression (accounting for non-VTE related death as a competing event), unadjusted and adjusted for TPX and study site [37]. To define an optimal cutoff for objective immobility, we will assess sensitivity and specificity at different mobility levels using time-dependent ROC-curve analysis accounting for censored data and competing events. To compare the predictive performance of the simplified Geneva score using the standard subjectively-assessed definition of immobilization (i.e. physician perception, Table 2) versus using objective accelerometry-assessed mobility measures, we will use likelihood ratio tests and/or the Akaike information criterion (AIC) as well as the c-statistics in competing risk models; mobility measures will be used as single covariates, as well as incorporated in the simplified Geneva score. We will assess the net

reclassification index to assess improvement in risk prediction if using the accelerometry-based mobility measure instead of the subjective immobility assessment for the simplified Geneva score.

## Expected impact and strengths

RISE will provide the first prospective head-to-head comparison of validated VTE RAMs. Previous studies suggest that pharmacological VTE prophylaxis is inappropriately used in medical patients: an international cross-sectional study reported that only about 40% of medical patients at high risk of VTE received appropriate prophylaxis, while on the other hand, it was inappropriately prescribed in half of all low risk patients [9,56,57]. Multiple reasons have been postulated for inadequate use of VTE prophylaxis in hospitalized medical patients, including the challenge to assess thromboembolic risk [58]. Our results will provide a clearer guidance for physicians about optimal VTE risk assessment and thus have the potential to facilitate and improve VTE prevention and reduce hospital-acquired VTE and associated deaths in medical inpatients. The simplified identification of patients who may really benefit from TPX may thus not only result in improved quality of care, but also in cost-savings.

A prospective cohort design is the optimal study design and provides the highest quality data to meet the aim of this study. This design will also correct the inherent limitations of the already published retrospective head-to-head VTE RAM comparison. A longitudinal study design is necessary to investigate prognostic measures, and prospective data collection allows complete and standardized measurements of exposures prior to the occurrence of any outcomes; also, objective mobility measurement is only possible in a prospective manner. Moreover, given the broad eligibility criteria of RISE, the results of this study will be generalizable to the population of hospitalized medical patients at risk of hospital-acquired VTE, i.e. those without intake of therapeutic anticoagulation.

As reported in previous studies, the subjective evaluation of patient's mobility is complex and unreliable [21,59,60]. In recent years, accelerometry-assessed mobility has become recognized as a valid and precise method to assess the mobility of inpatients [22–25]. A randomized Danish trial studying the effect of physical therapy on patient-reported outcomes after acute PE described several limitations using the incremental shuttle walk test as an objective mobility measure [59,61]. To this day, objective measures of mobility using accelerometry have only been assessed in studies with limited sample sizes [60,62]. RISE will be, to our knowledge, the first and the largest cohort studying VTE risk using accelerometry data.

Finally, the RISE cohort including 1350 general medical inpatients will be a valuable source for several secondary analyses, such as evaluating the association between TPX and bleeding, prospectively validating the IMPROVE bleeding risk score, and correlating nurse estimates of patients' mobility, using the Braden score, with objective measurements.

Thus, RISE has the potential to generate important knowledge about VTE prevention and risk stratification and to improve the quality of care of medical hospitalized patients.

## Supporting information

**S1 File. Study protocol and synopsis submitted and accepted by the Ethics committee.**
(PDF)

**S2 File. Approval letter of the Ethics committee of Bern (in German).**
(PDF)

**S3 File. Patient information sheet and informed consent (in German).**
(PDF)

**S4 File. Case-report form (baseline, discharge, follow-up at 90 days).**
(PDF)

**S5 File. Adjudication criteria.**
(PDF)

**S6 File. Completed SPIRIT checklist.**
(PDF)

## Acknowledgments

The authors want to thank all patients that have already agreed to participate in this study. We would also like to thank the study personnel that have recruited all the participants so far and the medical experts that have adjudicated all the already reported medical outcomes.

## Author Contributions

**Conceptualization:** Christophe Marti, Drahomir Aujesky, Andreas Limacher, Christine Baumgartner, Marie Méan.

**Data curation:** Damien Choffat, Pauline Darbellay Farhoumand, Damiana Rakovic, Christine Baumgartner, Marie Méan.

**Formal analysis:** Andreas Limacher, Jean-Benoît Rossel.

**Funding acquisition:** Christine Baumgartner, Marie Méan.

**Investigation:** Damien Choffat, Pauline Darbellay Farhoumand, Damiana Rakovic, Christine Baumgartner, Marie Méan.

**Methodology:** Damien Choffat, Christine Baumgartner, Marie Méan.

**Project administration:** Damien Choffat, Pauline Darbellay Farhoumand, Drahomir Aujesky, Damiana Rakovic, Christine Baumgartner, Marie Méan.

**Resources:** Damien Choffat, Pauline Darbellay Farhoumand, Christine Baumgartner, Marie Méan.

**Supervision:** Pauline Darbellay Farhoumand, Christine Baumgartner, Marie Méan.

**Validation:** Evrim Jaccard, Roxane de la Harpe, Vanessa Kraege, Malik Benmachiche, Christel Gerber, Salomé Leuzinger, Clara Podmore, Minh Khoa Truong, Céline Dumans-Louis, Christine Baumgartner, Marie Méan.

**Writing – original draft:** Damien Choffat, Jean-Benoît Rossel, Christine Baumgartner, Marie Méan.

**Writing – review & editing:** Damien Choffat, Pauline Darbellay Farhoumand, Evrim Jaccard, Roxane de la Harpe, Vanessa Kraege, Malik Benmachiche, Christel Gerber, Salomé Leuzinger, Clara Podmore, Minh Khoa Truong, Céline Dumans-Louis, Christophe Marti, Jean-Luc Reny, Drahomir Aujesky, Damiana Rakovic, Andreas Limacher, Jean-Benoît Rossel, Christine Baumgartner, Marie Méan.

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
