## [Decision Letter · Decision Letter 0]

14 Mar 2022

PONE-D-21-33782RIsk Stratification for hospital-acquired venous thromboEmbolism in medical patients (RISE): protocol for a prospective cohort studyPLOS ONE

Dear Dr. Choffat,

Thank you for submitting your manuscript to PLOS ONE. After careful consideration, we feel that it has merit but does not fully meet PLOS ONE’s publication criteria as it currently stands. Therefore, we invite you to submit a revised version of the manuscript that addresses the points raised during the review process.

Please submit your revised manuscript within Apr 25 2022 11:59PM. If you will need more time than this to complete your revisions, please reply to this message or contact the journal office at plosone@plos.org. Please include the following items when submitting your revised manuscript:A rebuttal letter that responds to each point raised by the academic editor and reviewer(s). You should upload this letter as a separate file labeled 'Response to Reviewers'.A marked-up copy of your manuscript that highlights changes made to the original version. You should upload this as a separate file labeled 'Revised Manuscript with Track Changes'.An unmarked version of your revised paper without tracked changes. You should upload this as a separate file labeled 'Manuscript'.If applicable, we recommend that you deposit your laboratory protocols in protocols.io to enhance the reproducibility of your results. Protocols.io assigns your protocol its own identifier (DOI) so that it can be cited independently in the future. For instructions see: https://journals.plos.org/plosone/s/submission-guidelines#loc-laboratory-protocols. Additionally, PLOS ONE offers an option for publishing peer-reviewed Lab Protocol articles, which describe protocols hosted on protocols.io. Read more information on sharing protocols at https://plos.org/protocols?utm_medium=editorial-email&utm_source=authorletters&utm_campaign=protocols.

We look forward to receiving your revised manuscript.

Kind regards,

Roza Chaireti

Academic Editor

PLOS ONE

Journal Requirements:

2. Thank you for stating the following in the Acknowledgments/ Funding Section of your manuscript: 

The RISE cohort is funded by several non-profit foundations (SGAIM Foundation, Novartis Biomedical Research Foundation, Swiss Heart Foundation, Chuard Schmidt Foundation, Gottfried und Julia Bangerter Foundation). The funding sources had no role in study design; in the collection, analysis and interpretation of data; in the writing of the report; and in the decision to submit the article for publication.

The RISE cohort is funded by several non-profit foundations (SGAIM Foundation, Novartis Biomedical Research Foundation, Swiss Heart Foundation, Chuard Schmidt Foundation, Gottfried und Julia Bangerter Foundation). The funding sources had no role in study design; in the collection, analysis and interpretation of data; in the writing of the report; and in the decision to submit the article for publication.

Reviewers' comments:

Reviewer's Responses to Questions

**Comments to the Author**

1. Does the manuscript provide a valid rationale for the proposed study, with clearly identified and justified research questions?

Reviewer #1: Yes

Reviewer #2: Yes

2. Is the protocol technically sound and planned in a manner that will lead to a meaningful outcome and allow testing the stated hypotheses?

Reviewer #1: Yes

Reviewer #2: Yes

3. Is the methodology feasible and described in sufficient detail to allow the work to be replicable?

Reviewer #1: Yes

Reviewer #2: Yes

4. Have the authors described where all data underlying the findings will be made available when the study is complete?

Reviewer #1: No

Reviewer #2: No

5. Is the manuscript presented in an intelligible fashion and written in standard English?

Reviewer #1: Yes

Reviewer #2: Yes

6. Review Comments to the Author

You may also provide optional suggestions and comments to authors that they might find helpful in planning their study.

Reviewer #1: General comments: The authors presented a protocol for a prospective cohort study on risk stratification for hospital-acquired venous thromboembolism in medical patients. Venous stasis as a predisposing factor for VTE in medical patients has been underrecognized, and this RCT will try to shed light on this aspect using objectively measured immobilization.

Specific comments:

1. Abstract, methods: you state “are calculated for each RAM”, but you have not listed or mentioned them. Essentially the scope of this study is to provide an objective assessment and grading of patient immobility, and assess its relative contribution in VTE risk, and eventually incorporate it in existing RAMs.

2. Page 3, last paragraph: Usually physician mobility orders are used and determine patient mobility; please comment.

3. Page 4, first paragraph: you mention the second aim of your study without mentioning its first aim.

Reviewer #2: In this study authors sent the protocol for a prospective cohort study

The aim of the study was prospectively validate the simplified Geneva score and to examine the predictive performance of a novel and objective definition of in-hospital immobilization using accelerometry.

- Primary objective is to prospectively validate the simplified Geneva score and to compare its prognostic performance with previously validated RAMs (i.e., the original Geneva, Padua, and IMPROVE scores).

- Second objective is to develop a new, objective, definition of inpatient immobilization using accelerometry and to compare its performance in predicting hospital acquired VTE with that of the subjective measurement.

The work is well written, and the study design is quite robust, however, I would like to make a couple of comments/suggestions:

In the estimation of the sample size, “a 90-day incidence of hospital-acquired VTE of respectively 2.8% and 0.6%, in high risk and low risk patients respectively, has been taken into account”. In this sense, the percentages shown refer to percentages of VTE diagnosed with or without thromboprophylaxis? Has the number of patients who will receive thromboprophylaxis been taken into consideration? It is true that the authors comment that a sensitivity analysis adjusted for the use of thromboprophylaxis will be performed, but if this situation has not been previously taken into account, it could influence the statistical power.

Page 10: the authors state that: the objective diagnosis of PE is based on available radiographic or autopsy reports (39-42). I don't know if I have come to understand this sentence, but the diagnosis of PE should be based on a CT-scan /angioCT

7. PLOS authors have the option to publish the peer review history of their article (what does this mean?). If published, this will include your full peer review and any attached files.

Reviewer #1: No

Reviewer #2: No

---

## [Author Response · Author response to Decision Letter 0]

9 Apr 2022

Roza Chaireti

Academic Editor

PLOS ONE

Lausanne, April 5th, 2022

Re: Revision of the manuscript entitled “RIsk Stratification for hospital-acquired venous thromboEmbolism in medical patients (RISE): protocol for a prospective cohort study.”

Dear editors,

Thank you for the opportunity to revise and resubmit our manuscript entitled “RIsk Stratification for hospital-acquired venous thromboEmbolism in medical patients (RISE): protocol for a prospective cohort study.” 

We have provided a point-by-point response to each comment below. 

Each author has approved the revised version, and we hope that these changes meet with your approval. 

Please contact Dr. Choffat at damien.choffat@chuv.ch with any remaining questions or for additional clarification. 

Thank you again for your time and consideration.

Sincerely,

Dr Damien Choffat

 

We ensured that our manuscript met the PLOS ONE’s style requirements.

2. Thank you for stating the following in the Acknowledgments/ Funding Section of your manuscript: The RISE cohort is funded by several non-profit foundations (SGAIM Foundation, Novartis Biomedical Research Foundation, Swiss Heart Foundation, Chuard Schmidt Foundation, Gottfried und Julia Bangerter Foundation). The funding sources had no role in study design; in the collection, analysis and interpretation of data; in the writing of the report; and in the decision to submit the article for publication. Please note that funding information should not appear in the Acknowledgments section or other areas of your manuscript. We will only publish funding information present in the Funding Statement section of the online submission form. Please remove any funding-related text from the manuscript and let us know how you would like to update your Funding Statement. Currently, your Funding Statement reads as follows: The RISE cohort is funded by several non-profit foundations (SGAIM Foundation, Novartis Biomedical Research Foundation, Swiss Heart Foundation, Chuard Schmidt Foundation, Gottfried und Julia Bangerter Foundation). The funding sources had no role in study design; in the collection, analysis and interpretation of data; in the writing of the report; and in the decision to submit the article for publication. Please include your amended statements within your cover letter; we will change the online submission form on your behalf.

We removed any funding information from the manuscript. Our Funding Statement section of the online submission is correct.

We confirm that our ethics statement is stated in the methods section.

We have reviewed the whole bibliography and can confirm that no cited article has been retracted. Furthermore, our bibliography list is complete and correct.

5. Does the manuscript provide a valid rationale for the proposed study, with clearly identified and justified research questions? The research question outlined is expected to address a valid academic problem or topic and contribute to the base of knowledge in the field.

a. Reviewer #1: Yes

b. Reviewer #2: Yes

Thank you very much for your remark.

6. Is the protocol technically sound and planned in a manner that will lead to a meaningful outcome and allow testing the stated hypotheses?

a. Reviewer #1: Yes

b. Reviewer #2: Yes

Thank you.

7. Is the methodology feasible and described in sufficient detail to allow the work to be replicable?

a. Reviewer #1: Yes

b. Reviewer #2: Yes

Thank you very much.

8. Have the authors described where all data underlying the findings will be made available when the study is complete?

a. Reviewer #1: No

b. Reviewer #2: No

This manuscript describes a protocol for a prospective cohort study, which is still ongoing. There are no datasets analyzed for this manuscript. Therefore, we cannot make the data fully available. 

9. Is the manuscript presented in an intelligible fashion and written in standard English?

a. Reviewer #1: Yes

b. Reviewer #2: Yes

Thank you very much.

10. PLOS authors have the option to publish the peer review history of their article (what does this mean?). If published, this will include your full peer review and any attached files. Do you want your identity to be public for this peer review? For information about this choice, including consent withdrawal, please see our Privacy Policy.

a. Reviewer #1: No

b. Reviewer #2: No

 

Reviewer #1: 

1. The authors presented a protocol for a prospective cohort study on risk stratification for hospital-acquired venous thromboembolism in medical patients. Venous stasis as a predisposing factor for VTE in medical patients has been under recognized, and this RCT will try to shed light on this aspect using objectively measured immobilization.

Thank you very much for your comment.

2. Abstract, methods: you state “are calculated for each RAM”, but you have not listed or mentioned them. Essentially the scope of this study is to provide an objective assessment and grading of patient immobility, and assess its relative contribution in VTE risk, and eventually incorporate it in existing RAMs.

Thank you for this remark. Indeed, we initially didn’t specify the name of the RAMs. We have now added the name of all the RAMs we will compare against the simplified Geneva score. 

Manuscript modifications (abstract, page 2): 

• Time-dependent area under the curve, sensitivity, specificity, and positive and negative predictive values are calculated for each RAM (i.e. the original Geneva score, Padua, IMPROVE score and simplified Geneva score) with and without the objective mobility measures to assess their accuracy in predicting hospital-acquired VTE at 90 days.

3. Page 3, last paragraph: Usually physician mobility orders are used and determine patient mobility; please comment.

Thank you for this interesting remark. Indeed, many hospitalized patients aren’t active during hospitalization even if they can move freely as shown by this recent study (reference: Tasheva P, et al. Accelerometry assessed physical activity of older adults hospitalized with acute medical illness - an observational study. BMC Geriatr. 2020 Oct 2;20(1):382). Physicians’ mobility orders are not systematically followed by hospitalized patients (please see our manuscript reference #21)

Therefore, we decided not to use physician mobility orders, which represent an ideal goal for patients that is then applied or not by nursing staff and patients themselves, but rather how physicians estimate their patients’ ability to move (i,e, their subjective estimation). Furthermore, we were interested in the physicians’ subjective mobility estimation since it is a key factor in the decision of implementing or not thromboprophylaxis. We now clarified this in the introduction section on page 3.

Manuscript modifications (introduction, page 3): 

• Patients and hospital staff also interpret physicians’ orders of mobilization with a substantial variation; for example, ambulation orders “out of bed to chair” can lead to a daily step count of 0 to 1800 (0-1.3 km) [21]. 

 

4. Page 4, first paragraph: you mention the second aim of your study without mentioning its first aim.

Thank you for your remark. Indeed, we understand how our phrasing was confusing. We therefore modified our manuscript. 

Manuscript modifications (introduction, page 3): 

• Therefore, we aim to establish the predictive performance of a novel and objective definition of in-hospital immobilization using accelerometry.

In addition, the aims of the study are mentioned as follows in the Objectives and hypotheses section on page 4: 

• “The primary objective is to prospectively validate the simplified Geneva score and to compare its prognostic performance with previously validated RAMs (i.e., the original Geneva, Padua, and IMPROVE scores). Therefore, we hypothesize that the novel, easier-to-use simplified Geneva score will be able to accurately detect medical inpatients at risk of hospital-acquired VTE and that it will be at least as accurate as previously validated RAMs. Our second objective is to develop a new, objective, definition of inpatient immobilization using accelerometry and to compare its performance in predicting hospital-acquired VTE with that of the subjective measurement. Accordingly, we hypothesize that objective, accelerometry-assessed mobility will be more accurate in predicting the risk of hospital-acquired VTE than subjective physician perception and that its incorporation into the simplified Geneva score will improve its prognostic performance.”

 

Reviewer #2: 

1. In this study authors sent the protocol for a prospective cohort study. The aim of the study was prospectively validate the simplified Geneva score and to examine the predictive performance of a novel and objective definition of in-hospital immobilization using accelerometry. Primary objective is to prospectively validate the simplified Geneva score and to compare its prognostic performance with previously validated RAMs (i.e., the original Geneva, Padua, and IMPROVE scores). Second objective is to develop a new, objective, definition of inpatient immobilization using accelerometry and to compare its performance in predicting hospital acquired VTE with that of the subjective measurement. The work is well written, and the study design is quite robust, however, I would like to make a couple of comments/suggestions:

Thank you very much for your comment.

2. In the estimation of the sample size, “a 90-day incidence of hospital-acquired VTE of respectively 2.8% and 0.6%, in high risk and low risk patients respectively, has been taken into account”. In this sense, the percentages shown refer to percentages of VTE diagnosed with or without thromboprophylaxis? Has the number of patients who will receive thromboprophylaxis been taken into consideration? It is true that the authors comment that a sensitivity analysis adjusted for the use of thromboprophylaxis will be performed, but if this situation has not been previously taken into account, it could influence the statistical power.

Thank you for this important comment. The incidence of VTE is calculated taking into account the provision of thromboprophylaxis. For clearer comprehension, we modified the sentence on page 12 as follows: 

“Based on a 2010 Swiss cohort study (13), we assume that that 67% of patients are categorized as high risk and 33% as low risk based on the simplified Geneva score and a 90-day incidence of hospital-acquired VTE with adequate thromboprophylaxis of 2.8% and 0.6%, in high risk and low risk patients, respectively, according to the simplified Geneva score (13). Therefore, we will need 1308 patients to detect an absolute risk difference of 2.2% between high and low risk patients, with a power of 80% at a 2-sided alpha of 0.05”

3. Page 10: the authors state that: the objective diagnosis of PE is based on available radiographic or autopsy reports (39-42). I don't know if I have come to understand this sentence, but the diagnosis of PE should be based on a CT-scan /angioCT

Indeed, we agree with the reviewer that this is confusing. We now modified the description on page 10: 

“[…] the objective diagnostic of PE is based on available radiology (CT pulmonary angiography, pulmonary angiography, or ventilation-perfusion lung scan) or autopsy reports”.

---

## [Decision Letter · Decision Letter 1]

10 May 2022

RIsk Stratification for hospital-acquired venous thromboEmbolism in medical patients (RISE): protocol for a prospective cohort study

PONE-D-21-33782R1

Dear Dr. Choffat,

We’re pleased to inform you that your manuscript has been judged scientifically suitable for publication and will be formally accepted for publication once it meets all outstanding technical requirements.

Kind regards,

Roza Chaireti

Academic Editor

PLOS ONE

Additional Editor Comments (optional):

Reviewers' comments:

Reviewer's Responses to Questions

**Comments to the Author**

1. Does the manuscript provide a valid rationale for the proposed study, with clearly identified and justified research questions?

Reviewer #2: Yes

2. Is the protocol technically sound and planned in a manner that will lead to a meaningful outcome and allow testing the stated hypotheses?

Reviewer #2: Yes

3. Is the methodology feasible and described in sufficient detail to allow the work to be replicable?

Reviewer #2: Yes

4. Have the authors described where all data underlying the findings will be made available when the study is complete?

Reviewer #2: Yes

5. Is the manuscript presented in an intelligible fashion and written in standard English?

Reviewer #2: Yes

6. Review Comments to the Author

You may also provide optional suggestions and comments to authors that they might find helpful in planning their study.

Reviewer #2: Authors have appropriately addressed all comments / suggestions.

No other comments

Congratulations

7. PLOS authors have the option to publish the peer review history of their article (what does this mean?). If published, this will include your full peer review and any attached files.

Reviewer #2: No

---

## [Editor Report · Acceptance letter]

16 May 2022

PONE-D-21-33782R1 

RIsk Stratification for hospital-acquired venous thromboEmbolism in medical patients (RISE): protocol for a prospective cohort study 

Dear Dr. Choffat:

I'm pleased to inform you that your manuscript has been deemed suitable for publication in PLOS ONE. Congratulations! Your manuscript is now with our production department. 

Kind regards, 

on behalf of

Dr. Roza Chaireti 

Academic Editor

PLOS ONE